# Monolithic back-end-of-line integration of phase change materials into foundry-manufactured silicon photonics

Maoliang Wei[1,7], Kai Xu[1,7], Bo Tang[2,7], Junying Li[1,3] ✉, Yiting Yun[1], Peng Zhang[2], Yingchun Wu[4,5], Kangjian Bao[4,5], Kunhao Lei[1], Zequn Chen[4,5], Hui Ma[1], Chunlei Sun ●[4,5], Ruonan Liu[2], Ming Li ●[6] ✉, Lan Li ●[4,5] ✉ & Hongtao Lin ●[1] ✉

Monolithic integration of novel materials without modifying the existing photonic component library is crucial to advancing heterogeneous silicon photonic integrated circuits. Here we show the introduction of a silicon nitride etch stop layer at select areas, coupled with low-loss oxide trench, enabling incorporation of functional materials without compromising foundry-verified device reliability. As an illustration, two distinct chalcogenide phase change materials (PCMs) with remarkable nonvolatile modulation capabilities, namely $Sb_2Se_3$ and $Ge_2Sb_2Se_4Te_1$, were monolithic back-end-of-line integrated, offering compact phase and intensity tuning units with zero-static power consumption. By employing these building blocks, the phase error of a push–pull Mach–Zehnder interferometer optical switch could be reduced with a 48% peak power consumption reduction. Mirco-ring filters with >5-bit wavelength selective intensity modulation and waveguide-based >7-bit intensity-modulation broadband attenuators could also be achieved. This foundry-compatible platform could open up the possibility of integrating other excellent optoelectronic materials into future silicon photonic process design kits.

Silicon photonics have broad prospects for practical applications in high-speed optical communication[1], microwave photonics[2], optical neural networks[3], and optical quantum computing[4]. Accurate modulation of the waveguide refractive index and absorption is critical for effectively implementing functional units in silicon photonics, facilitating precise control and adjustment of the functionality within silicon photonic networks[5]. However, current modulation schemes in photonic chips primarily rely on modulation approaches such as thermo-optic modulation and free-carrier dispersion in silicon, which exhibit a weak modulation strength (usually with an effective refractive index change $\Delta n_{eff} < 10^{-3}$) and necessitate a continuous power supply[6]. This leads to large device sizes and high static power consumption in photonic chips, limiting progress in large-scale optoelectronic integration.

Introducing innovative materials for functional units into silicon photonics has become imperative to attain exceptional device performance and reduce power consumption[7]. Various materials such as electro-optic polymer[8], metal-insulator-transition oxide[9], and 2D

[1]The State Key Lab of Brain-Machine Intelligence, Key Laboratory of Micro-Nano Electronics and Smart System of Zhejiang Province, College of Information Science and Electronic Engineering, Zhejiang University, Hangzhou 310027, China. [2]Institute of Microelectronics of the Chinese Academy of Sciences, Beijing 100029, China. [3]Hangzhou Institute for Advanced Study, University of Chinese Academy of Sciences, Hangzhou 310024, China. [4]Key Laboratory of 3D Micro/ Nano Fabrication and Characterization of Zhejiang Province, School of Engineering, Westlake University, Hangzhou, Zhejiang 310030, China. [5]Institute of Advanced Technology, Westlake Institute for Advanced Study, Hangzhou, Zhejiang 310024, China. [6]State Key Laboratory on Integrated Optoelectronics, Institute of Semiconductors, Chinese Academy of Sciences, Beijing 100083, China. [7]These authors contributed equally: Maoliang Wei, Kai Xu, Bo Tang. ✉e-mail: junyingli@zju.edu.cn; ml@semi.ac.cn; lilan@westlake.edu.cn; hometown@zju.edu.cn

materials[10], have been integrated for ultracompact or ultrafast volatile light modulation. The ongoing trend of integrating nonvolatile modulation materials such as charge-trapping materials[11,12], ferroelectric materials[13], and chalcogenide PCMs[14] is crucial for lowering the static power consumption of reconfigurable photonic circuits[15]. Despite the significant progress in prototype devices with exceptional performance, the fabrication process flows are incompatible with the existing silicon photonic foundry processes, rendering the established passive and active photonic component design kits unsuitable for direct application[16].

Enabling the monolithic integration of functional materials into silicon photonics while using the available process design kits (PDKs) is of utmost importance[15]. Chalcogenide PCMs, for example, can be directly deposited on silicon and have attracted significant attention because of their nonvolatile properties[17–20], making them promising candidates for compact (~10 μm) and zero-static power photonic devices. In recent decades, a plethora of PCM-integrated reconfigurable photonic devices have been extensively developed for intensity modulation[21–29], phase tuning[30–33], and light path switching[34,35]. Moreover, they play a crucial role in constructing photonic networks and serve as essential elements for optical storage[36], in-memory computing[37], and analog optical computing[38,39]. Despite the significant advancements in PCM-integrated photonics, realizing full compatibility when integrating PCMs into the entire silicon photonic fabrication flow remains highly challenging. The optoelectronic devices of PDKs inevitably rely on metal interconnections. Silicon waveguides are typically coated with more than 2-μm thick $SiO_2$ after the standard silicon photonic fabrication flow offered by foundries since a spacer layer between the silicon waveguide and the metal interconnections is necessary to avoid absorption loss of metal[40]. To achieve high-performance hybrid integrated silicon photonic devices with novel optoelectronic materials, such as PCMs, quantum dots[41], barium titanate ($BaTiO_3$, BTO)[42], lithium niobate ($LiNbO_3$)[43], electro-optic polymers, and van der Waals materials[44], minimizing the distance between the materials and silicon waveguides is crucial. However, the integration of excellent optoelectronic materials with the existing backend metals is challenging owing to the thick backend dielectric layer on top of the silicon waveguide. Therefore, an imperative back-end integration approach is urgently needed to facilitate post-processing integration of PCMs, thereby enhancing the feasibility of large-scale integration of nonvolatile reconfigurable optoelectronic chips.

In this paper, a foundry-compatible platform for monolithic back-end-of-line integration was demonstrated, facilitating large-scale integration of PCM-based photonic devices. By customizing the full process flow of silicon photonics and introducing a complementary metal–oxide–semiconductor (CMOS)-compatible silicon nitride (SiN) layer as an etch stop layer on Si waveguides, a deep $SiO_2$ trench with a low insertion loss of <0.09 dB/trench was etched for subsequent integration of various PCMs. Two kinds of chalcogenide PCMs, $Sb_2Se_3$ and $Ge_2Sb_2Se_4Te_1$ (GSS4T1), which have completely different nonvolatile modulation capabilities, were monolithically back-end-of-line integrated into silicon photonics. The electrical-assisted programmability of the fabricated devices was verified, enabling reconfigurable post-trimming and multilevel nonvolatile phase modulation and intensity modulation in optoelectronic chips. This endeavor not only showcased a back-end-of-line integration technique for combining PCMs with silicon photonics but also accomplished this goal without the need to modify the existing library of passive and active photonic components. Furthermore, this work provides a clear path for integrating other promising optoelectronic materials into future silicon optoelectronic chips.

## Results
### Back-end-of-line integration of PCMs into a commercial silicon photonic platform
Large-scale fabrication based on the CMOS platform without modifying the existing passive and active photonic component library is essential for realizing various practical applications of PCM-based nonvolatile electrically programmable photonic chips. However, PCMs are materials that are incompatible with standard CMOS processes. Therefore, back-end-of-line integration of PCMs is desirable for commercial photonics platform-compatible nonvolatile devices. Here, we propose a trench etching process utilizing SiN as the etch stop layer to realize deep $SiO_2$ cladding etching above functional areas (where the PCM would be deposited), which is suitable for developing CMOS back-end integration of multiple functional materials[45].

The SiN-assisted silicon photonic process was conducted on a 200-mm wafer at the IMECAS foundry, including low-loss $SiO_2$ trench etching above the functional areas. The integration of PCMs was accomplished by a back-end-of-line process involving ultraviolet lithography and film deposition, which exhibits significant potential for facilitating large-scale integration. The customized silicon photonic process incorporates a layer of SiN that serves as an etch stop layer to prevent damage to the Si waveguide during etching of $SiO_2$ trenches (see Fig. 1a). First, patterning of the photonic devices, implantation, and ion activation were implemented on a 200-mm silicon on insulator (SOI) wafer comprising a 220-nm silicon layer on top of a 2-μm buried oxide layer. Second, sequential deposition of 5-nm $SiO_2$ and 20-nm SiN was performed, followed by lithography and etching to fabricate the etch stop layer. Following the deposition of $SiO_2$ and subsequent metal interconnection, the silicon oxide above the waveguides in functional areas of the photonic devices was selectively etched. Due to the high etching selection ratio (>50:1) between $SiO_2$ and SiN, the etching process is effectively halted at the etch stop layer (SiN), thereby preventing damage to the silicon waveguide. The high selectivity between $SiO_2$ and SiN originates from the utilization of an etching gas with a high carbon–fluorine ratio ($C_4F_6$). Finally, a wet etching process was employed to remove the SiN layer above the functional areas.

Benefiting from the etch stop layer, waveguides with low-loss narrow $SiO_2$ trenches were realized, with their measured transmission spectra shown in Fig. 1b, where the numbers of cascaded trenches are 10, 20, and 30. The cut-back measurement suggested that the insertion loss (IL) introduced by a single trench is as low as 0.083 dB at 1550 nm, as shown in Fig. 1c. Low-loss trenches provide a convenient approach for back-end integration of PCMs, facilitating compatibility with verified silicon photonic devices in commercially available PDKs. To prevent any performance degradation of the device, the post-processing temperature should be below 450 °C[46]. Additionally, post-fabrication patterning was accomplished exclusively by ultraviolet lithography, thereby possessing the potential for large-scale integration with high throughput. A flowchart of the detailed fabrication process for back-end integration of PCMs is illustrated in Fig. 1a. A PCM thin film was deposited through magnetron sputtering followed by a lift-off process upon opening the $SiO_2$ cladding trench window (processing temperature <150 °C). Afterward, the chip was annealed at 200 °C and 300 °C in a nitrogen atmosphere for 15 minutes to facilitate crystallization of $Sb_2Se_3$ and $GeSbSe_4Te_1$, respectively. A 30 nm-thick $Al_2O_3$ protective layer was then deposited by atomic layer deposition (process temperature <150 °C). Finally, the contact window above the metal electrode was etched to ensure electrical interconnection (process temperature <110 °C). Figure 1d illustrates the functional region structure after fabrication. The electrothermal control of the PCM induced by the PIN diode enables a nonvolatile response of the effective refractive index in the hybrid waveguides.

### Back-end integrated $Sb_2Se_3$ enabled trimming of a push-pull Mach–Zehnder interferometer (MZI) switch
Silicon photonic devices inevitably suffer from fabrication errors and consequent deviation of the performance from the intended design. Moreover, the device-to-device deviations could be significantly magnified in networks, leading to increased complexity when configuring photonic integrated circuits (PICs). Post-fabrication trimming

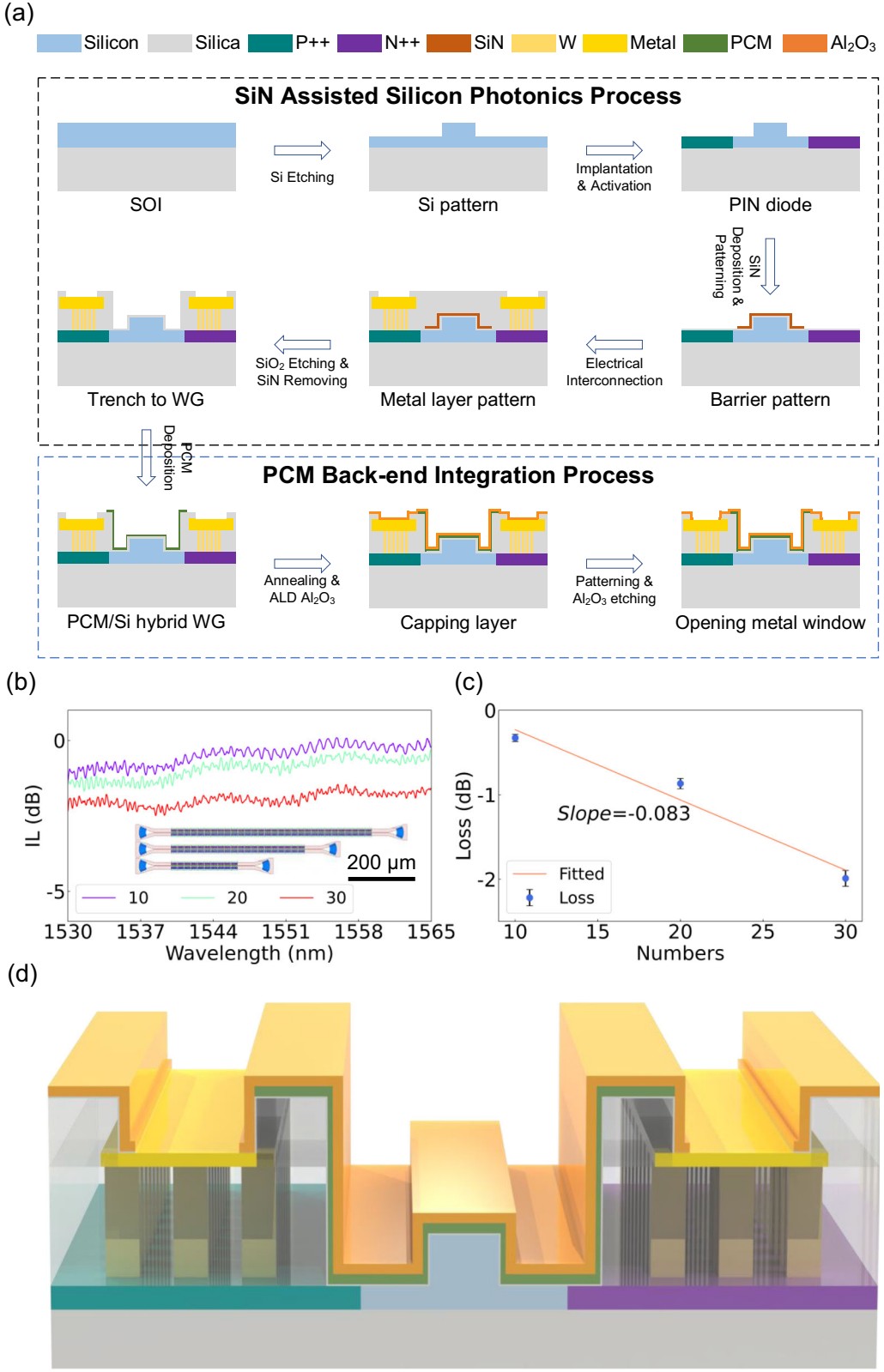

**Fig. 1 | SiN-assisted silicon photonic process for back-end-of-line integration of PCMs. a** Flowchart of the device fabrication process. **b** Measured transmission spectra of waveguides with different numbers of cascaded trenches. The inset shows the layout of the cascaded devices. **c** Waveguide loss introduced by the $SiO_2$ trench etching process assessed employing the cut-back method. The error bars corresponding to the standard deviation of multiple loss characterization results. **d** 3D schematic image of the device after fabrication.

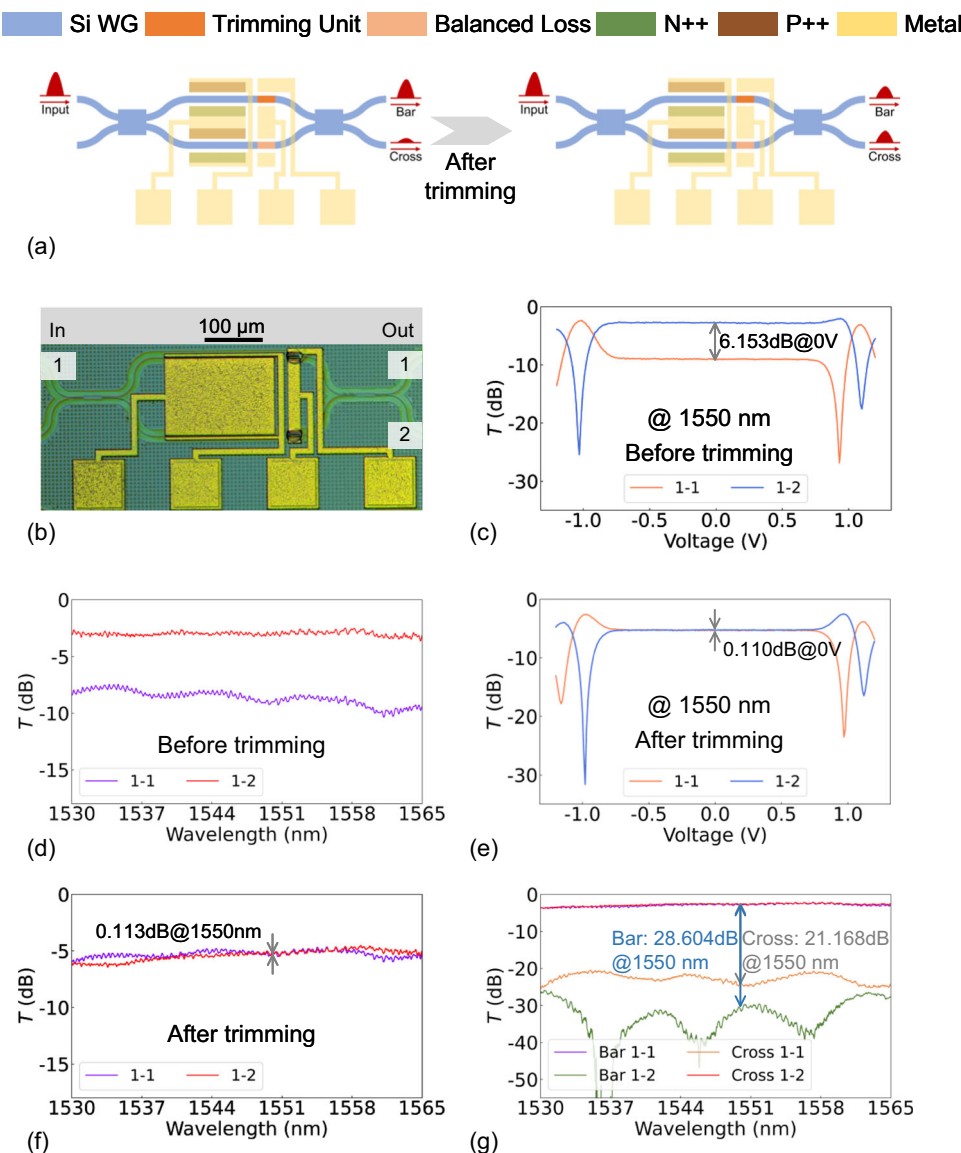

**Fig. 2 | Principle and performance of the push-pull MZI switch with a PFT capability. a** PFT principle for the push-pull MZI switch. The splitting ratio of the two output ports is balanced at 0 V through precise trimming. **b** Microscope image of the device. **c** Measured optical power–voltage (O-V) curves and **d** measured spectra before PFT. **e** O-V curves and **f** measured spectra after PFT. **g** Measured spectra for the bar and cross states after PFT.

(PFT) enables calibration of photonic devices after fabrication. Compared to other trimming methods, including femtosecond-laser annealing[47] and Ge ion implantation-and-annealing[48,49], active and reversible trimming enabled by PCM manipulation[30] possesses significantly improved degrees of freedom. In this section, we demonstrate a post-fabrication active trimming technique by electrically fine-tuning a low-loss $Sb_2Se_3$ patch integrated using back-end integration.

The schematic diagram in Fig. 2a illustrates the device structure and operational principle of the low-loss PCM-based PFT, exemplified by a push-pull MZI switch. A section of a p-i-n doped Si waveguide, covered with $Sb_2Se_3$ patch, was employed as a trimming unit. An identical structure was set on another branch of the MZI to balance the optical losses on the two arms. During the PFT, a sequence of electrical pulses was applied to the trimming unit, thereby triggering amorphization of $Sb_2Se_3$ and inducing a nonvolatile change in the refractive index of the $Sb_2Se_3$/Si hybrid waveguide. Therefore, the optical power between the two output ports gradually reached equilibrium, accompanied by a balance of the drive voltages.

A microscope image of the push-pull MZI is presented in Fig. 2b. Before PFT, as shown in Fig. 2c, the optical power-voltage (O-V) curve suggested that the optical power splitting ratio between the two output ports was greater than 6 dB at 0 V, and the drive voltage for the bar state (cross state) was −1.03 V (0.93 V), corresponding to a power consumption of 4.55 mW (1.00 mW). Additionally, a significant optical power imbalance spanning the C-band was observed, as shown in Fig. 2d. Therefore, the insertion losses of the bar and cross states exhibited significant disparities due to the inherent losses associated with carrier-injection-based phase modulation. (>0.4 dB at 1550 nm; the measured spectra can be found in Supplementary Fig. S1a).

After applying a series of electrical pulses with various amplitudes (ranging from 3 V to 6 V with an interval of 0.01 V) and a fixed pulse duration of 500 ns, precise control over the amorphous and crystalline mixing state of $Sb_2Se_3$ was achieved, thereby enabling device trimming without impacting the electrical properties of the device (Supplementary Fig. S1b). The imbalance between the two ports was reduced to 0.110 dB at 0 V, as depicted by the O-V curve in Fig. 2e. The drive

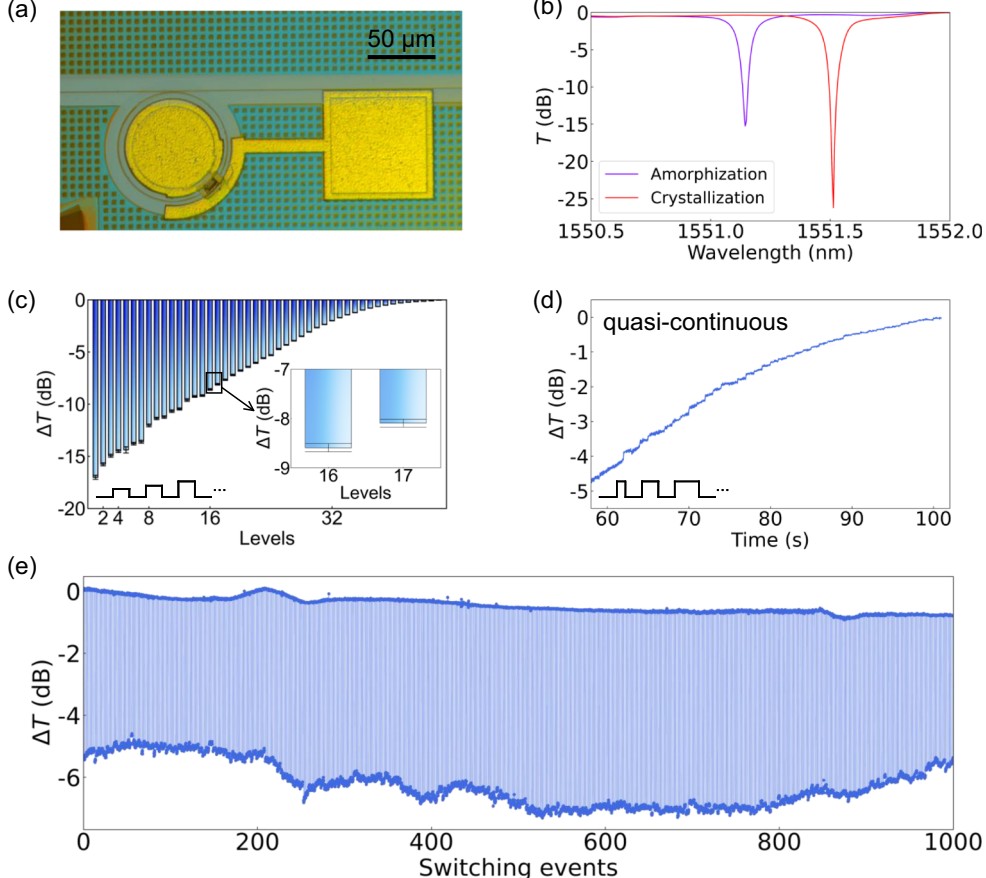

**Fig. 3 | Back-end integration of Sb₂Se₃ with an MRR for multilevel phase modulation. a** Microscope image of the MRR. **b** Measured spectra of the reversible switching events. **c** Multilevel crystallization process (ΔT at 1551.145 nm) induced by applying PVAFD. The inset shows the zoomed-in error bar of two distinct states, which corresponding to the standard deviation of multiple transmission values in each level. **d** Multilevel crystallization process (ΔT at 1551.145 nm) induced by applying PVDFA. **e** Five hundred reversible switching events of the device. The transmittance change (ΔT) is the difference in optical power at 1551.513 nm.

voltages of bar and cross states were reduced to −0.98 V (yielding a power consumption of 2.33 mW) and 0.97 V (yielding a power consumption of 2.01 mW), respectively. This thereby reduced the voltage disparity from 0.1 V to 0.01 V, and improved the total and peak power efficiencies by >20% and >48%, respectively. Moreover, the trimming unit effectively equalized the splitting ratio of the two output ports across the entire C-band (see Fig. 2f). The imbalance of the insertion losses in the bar (2.69 dB) and cross (2.62 dB) states was effectively minimized to a mere 0.07 dB at 1550 nm while simultaneously ensuring an extinction ratio >20 dB in both states (see Fig. 2g). The IL of the trimming unit was 0.46 dB, estimated using the cut-back method, as detailed in Supplementary Fig. S2. The characterization of the push-pull MZI without Sb₂Se₃ also confirmed that the primary source of IL was not the trimming unit (see Supplementary Fig. S3), but rather the propagation loss of the Si waveguide and the mode mismatch between the straight and bent sections. The trimmed MZI exhibited a high switching speed with 10-90% rise and 90-10% fall times of 9.26 ns and 9.81 ns, respectively (Supplementary Fig. S1c). Additionally, on-off switching (>10⁷) and storage at room temperature (12 days) had no significant impact on the optical performance of the device, indicating that the use of a Sb₂Se₃-based trimming unit is a reliable method for PFT (refer to Supplementary Fig. S1 for detailed measurement results).

The Sb₂Se₃-based trimming unit provides a reliable PFT technique, which not only achieves a balance of the drive voltages and insertion losses but also significantly reduces both the total and peak power consumption of a push-pull MZI. The embedded trimming units are essential for simplifying the control and reducing the power consumption of very large-scale PICs.

## Reconfigurable nonvolatile multilevel low-loss phase modulation

Compact nonvolatile multilevel phase modulation not only mitigates static power consumption but also enhances the integration density, making it a promising technique for constructing reconfigurable PICs such as those used in microwave photonics[50], quantum computing[51], and coherent optical computing[52]. Here, we demonstrated low-loss multilevel phase modulation using a back-end integrated Sb₂Se₃/Si hybrid waveguide in a microring resonator (MRR).

The nonvolatile MRR switch, featuring an 8 μm-long Sb₂Se₃/Si hybrid waveguide, is shown in Fig. 3a. An ~25-nm Sb₂Se₃ patch was adopted to mitigate the mode mismatch loss arising from the interface between the bare silicon waveguide and the Sb₂Se₃/Si hybrid waveguide (see Supplementary Fig. S4 for a detailed analysis of the loss induced by mode mismatch). Reversible switching was achieved by applying a 7.15 V/500 ns pulse for amorphization and a 2.25 V/100 ms pulse for crystallization, resulting in an extinction ratio (ER) larger than 25 dB at 1551.513 nm (see Fig. 3b). The observed change in the measured spectra indicates that a phase shift of ~0.3 π was achieved, accompanied by a crystallization-induced loss of 0.0223 dB/μm.

To achieve multilevel switching, two types of manipulating pulses were employed; electrical pulses with various amplitudes and a fixed duration (PVAFD) and pulses with various durations and a fixed amplitude (PVDFA). The change in transmittance (ΔT) at 1551.145 nm

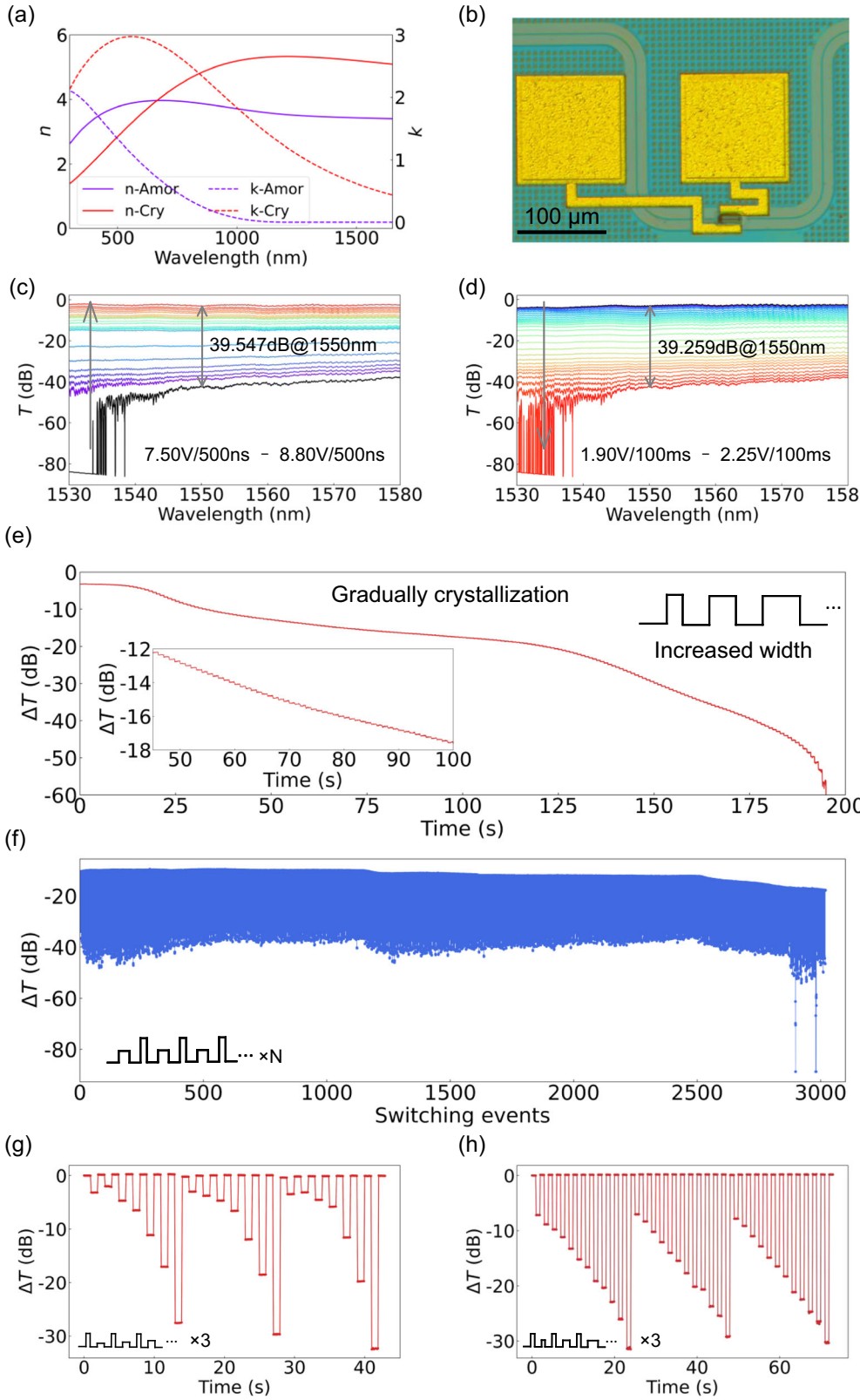

**Fig. 4 | Back-end integration of GSS4T1 with straight waveguide for multilevel intensity modulation. a** Optical constant of the GSS4T1 film on a silicon substrate before and after annealing. **b** Microscope image of the photonic attenuator. Measured spectra for multilevel amorphization (**c**) and crystallization (**d**). For amorphization, 500-ns pulses with amplitudes ranging from 7.50 to 8.80 V with an interval of 0.05 V were employed, excluding values of 7.55 V, 7.60 V and 7.65 V, but including 8.02 V. For crystallization, 100-ms pulses ranging in amplitude from 1.90 V to 2.25 with interval of 0.01 V were applied, excluding values of 2.21 V, 2.22 V, 2.23 V and 2.24 V. **e** Multilevel crystallization induced by applying PVDFA. The inset shows the enlarged gradual crystallization. **f** Fifteen hundred reversible switching events of the photonic attenuator. Arbitrary state configurations induced by various amplitudes (**g**) and durations (**h**). All the transmittance changes were all measured at 1550 nm.

was utilized to monitor the nonvolatile multilevel phase modulation exhibited by the MRR. As shown in Fig. 3c, 36-level (>5-bit) crystallization was realized by applying PVAFD with an interval of 0.01 V. However, achieving higher levels of nonvolatile switching using PVAFD with an amplitude interval smaller than 0.01 V poses a significant challenge. Hence, PVDFA exhibits the potential to accommodate more distinguishable states while inducing multilevel crystallization due to the presence of numerous pulses (nearly $10^5$ at a resolution of 1 μs) with durations <100 ms. Here, quasi-continuous switching was achieved by applying PVDFA with gradually increasing duration (see Fig. 3d and Supplementary Fig. S5). The demonstration of 500 switching events is also presented (see Fig. 3e).

The back-end integrated $Sb_2Se_3$/Si hybrid waveguide offers reversible low-loss phase modulation, enabling fine multilevel switching by employing PVDFA, providing an attractive fine-phase tuning solution for large-scale PCM-driven photonic networks.

### Reconfigurable nonvolatile multilevel intensity modulation

Intensity modulation has been widely used in optical computing[53], optical communication[54], and microwave photonics[55]. GSS4T1, possessing a low-loss amorphous state and a lossy crystalline state at telecom wavebands[19] has emerged as a promising candidate for nonvolatile intensity modulation compared with traditional GST materials. Here, we pioneered the demonstration of an electrically programmable waveguide-integrated broadband optical attenuator employing back-end integrated GSS4T1.

The back-end integrated photonic attenuator offers a high ER with a small footprint (16 μm) owing to the high extinction coefficient contrast ($\Delta k = 0.549$) between different states of our sputtered GSS4T1 films (see Fig. 4a). A microscope image of the fabricated device is shown in Fig. 4b. Reversible multilevel switching of the photonic attenuator was achieved by applying PVAFD, as shown in Fig. 4c, d. The IL and ER were measured to be 2.91 dB and 39.5 dB, respectively. Furthermore, multilevel intensity modulation exceeding 180 levels (>7 bits) was achieved by applying PVDFA (see Fig. 4e). After 1500 switching events, there was no obvious deterioration in the device performance (see Fig. 4f). Further switching events, both with and without an optimized coupling efficiency of the grating coupler after the cyclic measurement, suggested that the observed increase in optical loss could be attributed to a shift in the coupling state.

Although fine-tuning for any target state can always be accomplished via multiple fine-correction pulses akin to the trimming process, an arbitrary state configuration is essential in certain scenarios. An arbitrary state configuration is usually achieved by applying an amorphization pulse followed by a crystallization pulse. Through this process, we could realize a >3-bit arbitrary state configuration for the GSS4T1-based attenuator. By applying pulses with various amplitudes (various durations), a total of 7 (11) distinguishable states were achieved (see Fig. 4g, h). Achieving high resolution solely through the application of two pulses is challenging due to the inherent random nucleation during crystallization. To obtain a reliable arbitrary state configuration, combining fine-tuning with feedback monitoring is a promising solution. A reliable intermediate state can be achieved by applying a single pulse for amorphization, followed by multiple pulses for gradual crystallization. Although a specific state configuration requires multiple pulses, this approach is anticipated to achieve precise multilevel switching and even quasi-continuous configurations.

We have demonstrated a back-end integrated GSS4T1-based photonic attenuator with an ER > 39 dB and multilevel switching >7 bits, thereby forging a path for large-scale nonvolatile intensity-modulated PICs.

## Discussion

The integration of novel materials into existing passive and active photonic component libraries, enabling the incorporation of unprecedented devices, is crucial for developing next-generation heterogeneous silicon PICs. In this study, we presented a platform for monolithic back-end-of-line integration of PCMs into foundry-manufactured silicon photonics. Narrow $SiO_2$ trenches were successfully etched down to the top surface of the Si waveguide core with a SiN etch stop layer on it, enabling post-deposition of various PCMs for large-scale nonvolatile photonic device integration without any impact on the foundry-verified photonic devices. The IL of the customized trench was less than 0.09 dB, allowing large-scale integration into photonic networks.

Two kinds of chalcogenide PCMs, $Sb_2Se_3$ and $Ge_2Sb_2Se_4Te_1$, which have completely different nonvolatile modulation capabilities, were monolithically back-end-of-line integrated into silicon photonics. Nonvolatile post-trimming achieved with post-integrated $Sb_2Se_3$ was demonstrated. By electrically fine-tuning the trimming unit to achieve a balance between the push and pull voltages, the peak power consumption of a push-pull MZI-type switch was reduced by 48%. The nonvolatile reconfigurable PFT could be applied to various photonic chips to prevent fabrication errors. Next, both nonvolatile phase modulation based on $Sb_2Se_3$ and nonvolatile intensity modulation based on $Ge_2Sb_2Se_4Te_1$ employing back-end integration were demonstrated for large-scale nonvolatile programmable photonic networks. The $Sb_2Se_3$-integrated microring switch achieved >5-bit multilevel switching, exhibiting the potential for quasi-continuous switching with electric-pulse-width modulation. Moreover, the $Ge_2Sb_2Se_4Te_1$-integrated-waveguide-based broadband attenuator achieved a maximum ER > 39 dB with >7-bit multilevel modulation. The post-fabricated intensity modulator endured 1500 stable switching cycles without obvious performance degradation.

To the best of our knowledge, we have demonstrated, for the first time, monolithic back-end-of-line integration of PCMs with chips based on a commercial foundry process flow offered by a 200-mm commercial silicon photonic foundry and thus electrically programmable multilevel switching nonvolatile photonic devices. These findings highlight the feasibility of realizing large-scale-fabricated programmable PCM-based nonvolatile photonic chips, which hold significant potential for low-power, large-scale applications in optical computing, microwave photonics, and optical communication networks. The validated foundry-compatible, monolithic back-end-of-line integration platform will also pave the way for the integration of other excellent optoelectronic materials (including PCMs, quantum dots, BTO, $LiNbO_3$, electro-optic polymers, and van der Waals materials) into future silicon photonic PDKs.

## Methods

### Device characterization

A broadband tunable laser (Santec TSL-550) emitted a signal light that was directed to a polarization controller (PC) for polarization adjustment, and subsequently coupled with the device under test (DUT) through a double-end grating coupler. The output optical power of the DUT was measured by an optical power meter (MPM-210). An electric pulse was generated by an arbitrary waveform generator (SDG7052A) and applied to the electrode of the DUT through a radio frequency probe.

## Data availability

All the data supporting this study are available in the paper and Supplementary Information. Additional data related to this paper are available from the corresponding authors upon request.

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

## Acknowledgements

This work was supported by the National Natural Science Foundation of China (91950204 received by H.L., 62105287 received by J.L., 61975179 received by H.L., and 92150302 received by H.L.), the National Key Research, Development Program of China (2021YFB2801300 received by J.L. and 2019YFB2203002 received by H.L.), the Zhejiang Provincial Natural Science Foundation of China (LD22F040002 received by L.L.), the Key Project of Westlake Institute for Optoelectronics (110500Y0022303 received by L.L.). The authors would like to acknowledge the fabrication support from the Institute of Microelectronics of the Chinese Academy of Sciences, ZJU Micro-Nano Fabrication Center at Zhejiang University, and Westlake Center for Micro/Nano Fabrication at Westlake University. The authors would also like to thank Liming Shan for his help in chalcogenide thin-film deposition and Mengxue Qi for her help in device fabrication.

## Author contributions

H.L. conceived the idea. M.W., K.X. and B.T. carried out the fabrication, measurement setup construction, and device testing. J.L. contributed to the fabrication. Y.Y. assisted in device testing. P. Z. performed the passive device characterization. Y.W. performed the protective layer deposition. K.B. assisted in the wafer dicing. K.L. developed the PCM films. Z.C. developed and deposited the PCM films. H.M. assisted with the measurement setup construction. C.S. contributed to the passive device design. R.L. contributed to the tape-out of the GDS files. H.L., L.L., J.L. and M.L. supervised the research. All the authors contributed to the technical discussions and writing of the paper.

## Competing interests

The authors declare no competing interests.
