## [Peer Review File · Nature Communications]

Monolithic back-end-of-line integration of phase change materials into foundry-manufactured silicon photonicsREVIEWER COMMENTS

Reviewer #1 (Remarks to the Author):

The paper describes backend integration of PCMs with a silicon photonic foundry fabrication process and an array of reconfigurable photonic devices with exceptional performances. This is a paper that will likely be of significant interest to broad audiences in the photonics field given the rising importance of new active optical materials and reconfigurable photonics. I recommend publication of the manuscript provided that the following comments are adequately addressed.

1) The term "zero-change" is a misnomer. The process discussed here involves several steps that are, at best, implemented only at specific photonic foundries. For example, the conformal SiN layer coating on Si waveguides is by no means common practice. In the case where SiN etch stop is involved, most foundries use a SiN layer separated from the SOI layer in their standard process offering. Similarly, the oxide trench opening step is customarily offered as an add-on to the standard fabrication process of most photonic foundries. I would argue that the novelty of the paper in terms of fabrication process is not about that the authors stick to a plain vanilla foundry process, but that they demonstrated integration of PCMs in the presence of backend metal. I therefore suggest the author re-writing their introduction and discussion sections, removing references to "zero-change" and the like while emphasizing on the integration implementation through backend layers.

2) The resolution of several figures (Figs. 1, 2, and 4) is poor and the color contrast with the white background is low. I recommend using a different color to label the SOI waveguide in Fig. 1. Fig. 1 also implies that the PCM deposition is conformal over the trench sidewalls, which is likely not the case for sputtering and should be modified.

3) The method used to etch the SiO₂ window (with 50:1 selectivity to SiN) is not specified.

4) While the overall presentation of the paper is clear, there are still quite some grammatical errors that need to be fixed. Below is an inexact list. I strongly recommend the authors to hire some professional language editing service to polish the manuscript.

"a silicon nitride etching stop layer at selective area" -> "a silicon nitride etching stop layer at selective areas"

"Introducing innovative materials for function units" -> "Introducing innovative materials for functional units"

"An ongoing trend to integrating nonvolatile modulated materials" -> "An ongoing trend to integrating nonvolatile modulation materials"

"Therefore, a back-end-of-line integration" -> "Therefore, back-end-of-line integration"

"Secondly, a sequential deposition" -> "Secondly, sequential deposition"

"deposited through magnetron sputtering followed by a lift-off process upon the SiO₂ cladding trench window (process temperature <150 °C). Afterward" -> "deposited through magnetron sputtering followed by a lift-off process upon opening the SiO₂ cladding trench window (processing temperature <150 °C). Afterwards"

"to achieve optical loss equilibrium" -> "to balance optical losses on both arms"

Reviewer #2 (Remarks to the Author):

The authors report on a technique that enables the monolithic BOEL integration of phase change materials. This platform/technique is labelled as "Zero Change" as it is fully compatible with standard silicon photonic processes and does not require modifications in their PDKs. This is a very important topic as the hybrid integration of other photonic materials to silicon chips is essential to enable functionalities such as amplification, phase shifting and attenuation, to cite a few. The value here resides in that the FOEL process does not need to be changed.

In this particular paper the authors report a technique that is based on etching down SiO₂ narrow trenches on the top of Si waveguide cores with a SiN etching stop layer that enables the safe post-deposition of other photonic materials. This technique is original to my understanding. The authors have successfully deposited two different chalcogenide PCMs (Sb₂Se₃ and Ge₂Sb₂Se₄Te) and demonstrated the operation in three applications (post-trimming, phase and amplitude modulation).

The paper reports a relevant progress over the state of the art and my suggestions and comments for minor revisions are focused towards improving some specific parts.

- a) Regarding Fig 1(a) it is difficult to appreciate the limits of some of the regions. I believe this is because the colors that are chosen have little contrast. In particular I refer to the silicon and silica layers. Perhaps changing one of them can improve the visibility. Also in that figure the authors should specify the trenches limits
- b) When discussing the technique the authors indicate that a post-fabrication patterning will be required including annealing. How compatible would this be with a standard silicon photonics fabrication (i.e. would this be needed to be carried in a completely different facility?)
- c) The materials considered in this study are chalcogenide based and the authors mention that the technique can open the path to integrate "other excellent optoelectronic materials". However which materials are is not further discussed. Can the authors elaborate more on this topic?
- d) When dealing with the push-pull MZI switch an insertion loss of 0.46 dB is reported. This is a quite high figure, especially considering an application to programmable photonics where a target figure of 0.1-0.15 dB per gate is desired for scalable chips. Can the authors comment on possible ways to reduce such losses?
- e) In the case of the integration of SbSe in a MRR for multilevel phase modulation the authors report a phase shift of 0.3π. However for full functionality a 2π phase shift should be achieved. Can the authors comment?
- f) I have real problems in reading the figures as the fonts are very small. Perhaps the authors could consider reducing the number of subfigures in the main document and transfer some to the supplementary materials.
- g) A final discussion on reliability derived from the use of pulsed techniques and material transition from amorphous to crystalline would be useful to the reader.

Response to Reviewer Comments NCOMMS-23-47181-T

\P

REVIEWER COMMENTS

Reviewer #1 (Remarks to the Author):

The paper describes backend integration of PCMs with a silicon photonic foundry fabrication process and an array of reconfigurable photonic devices with exceptional performances. This is a paper that will likely be of significant interest to broad audiences in the photonics field given the rising importance of new active optical materials and reconfigurable photonics. I recommend publication of the manuscript provided that the following comments are adequately addressed.

1) The term "zero-change" is a misnomer. The process discussed here involve several steps that are, at best, implemented only at specific photonic foundries. For example, the conformal SiN layer coating on Si waveguides is by no means common practice. In the case where SiN etch stop is involved, most foundries use a SiN layer separated from the SOI layer in their standard process offering. Similarly, the oxide trench opening step is customarily offered as an add-on to the standard fabrication process of most photonic foundries. I would argue that the novelty of the paper in term of fabrication process is not about that the authors stick to a plain vanilla foundry process, but that they demonstrated integration of PCMs in the presence of backend metal. I therefore suggest the author re-writing their introduction and discussion sections, removing references to "zero-change" and the like while emphasizing on the integration implementation through backend layers.

Response:

Many thanks to the reviewer for the helpful comment. As noted by the reviewer, the novelty of our work is the integration of PCM in the presence of backend metal by slightly changing the fabrication flow, thereby circumventing changes to existing PDKs. With this scheme, we demonstrated several applications including post-trimming, phase modulation and amplitude modulation, by the backend integration of PCMs (Sb_2Se_3 and GSS_4T_1). On one hand, in our previous manuscript, we have added quotes to the zero change statement to indicate that it is not truly a zero change, due to the modified fabrication flow implemented for achieving low-loss trench. On the other hand, an etch stop layer (SiN) was deposited only on top of the functional areas, thereby ensuring a negligible impact on the performance of other silicon photonic devices. Therefore, another purpose of the "zero change" we previously proposed is to emphasize that this scheme can achieve the integration of the PCM without changing the existing PDKs.

Based on the reviewer's comment, to emphasize the novelty of our work, we have changed the expression in our revised manuscript, from "zero-change" to "near-zero-change". Moreover, we have made modifications to the title, introduction, and conclusion of our revised manuscript, emphasizing the backend integration of the PCM rather than focusing on the "zero change" platform. The modified content is as follows:

For the title:

“Near-zero-change” platform for monolithic back-end-of-line integration of phase change materials into silicon photonics

For the introduction:

Enabling the monolithic integration of functional materials into silicon photonics while using the available process design kits (PDKs) is of utmost importance.

The optoelectronic devices of PDKs inevitably rely on metal interconnections. Silicon waveguides are typically coated with more than 2- μm thick SiO_2 after the standard silicon photonic fabrication flow offered by foundries since a spacer layer between the silicon waveguide and the metal interconnections is necessary to avoid absorption loss of metal. To achieve high-performance hybrid integrated silicon photonic devices with novel optoelectronic materials, such as PCMs, quantum dot materials, and electro-optic polymers, minimizing the distance between the materials and silicon waveguides is crucial. However, the integration of excellent optoelectronic materials with the existing backend metals is challenging owing to the thick backend dielectric layer on top of the silicon waveguide.

In this paper, a “near zero-change” platform for a monolithic back-end-of-line integration scheme facilitating the large-scale integration of PCM-based photonic devices was demonstrated.

For the conclusion:

The “near-zero-change” monolithic back-end-of-line integration platform could also pave the way for the integration of other excellent optoelectronic materials (including PCMs, quantum dot materials, and electro-optic polymers) into future silicon photonic PDKs.

2) The resolution of several figures (Figs. 1, 2, and 4) is poor and the color contrast with the white background is low. I recommend using a different color to label the SOI waveguide in Fig. 1. Fig. 1 also implies that the PCM deposition is conformal over the trench sidewalls, which is likely not the case for sputtering and should be modified.

Response:

We sincerely appreciate the reviewer for the insightful comment. We apologize for the suboptimal resolution and color contrast of our figure. To address these issues, we have carefully revised our figures. Moreover, the non-conformal deposition of PCMs is considered in our revised flowchart of the fabrication process. In the revised fabrication flowchart, the PCM thickness at the sidewall of the waveguide is thinner than that above the waveguide. The revised figures have been updated in our revised manuscript, as follows:

For Fig.1

For Fig.2

Si WG Trimming Unit Balanced Loss N++ P++ Metal

For Fig.4

3) The method used to etch the SiO₂ window (with 50:1 selectivity to SiN) is not specified.

Response:

Many thanks to the reviewer for the comment. The achievement of high selectivity between SiO₂ and SiN was facilitated by using a high-carbon-fluorine ratio etching gas (C₄F₆). During the SiO₂ etching process, oxygen (O) reacts with carbon (C) to form either CO or CO₂, which subsequently evaporates. For the etching of SiN, no element can react with C to generate gas, thereby leading to the formation of a polymer layer on the surface as a protective barrier against SiN etching. Therefore, SiN can serve as an effective etching stop layer during the process of SiO₂ etching. We are sorry that we are unable to disclose specific process parameters due to the confidentiality requirements associated with the pilot CMOS line. To elucidate the method, we have incorporated the etching gas into our revised manuscript as follows:

The high selectivity between SiO₂ and SiN originates from the utilization of an etching gas with a high carbon–fluorine ratio (C₄F₆).

4) While the overall presentation of the paper is clear, there are still quite some grammatic errors that need to be fixed. Below is an inexhaustive list. I strongly recommend the authors to hire some professional language editing service to polish the manuscript.

"a silicon nitride etching stop layer at selective area" -> "a silicon nitride etching stop layer at selective areas"

"Introducing innovative materials for function units" -> "Introducing innovative materials for functional units"

"An ongoing trend to integrating nonvolatile modulated materials" -> "An ongoing trend to integrating nonvolatile modulation materials"

"Therefore, a back-end-of-line integration" -> "Therefore, back-end-of-line integration"

"Secondly, a sequential deposition" -> "Secondly, sequential deposition"

"deposited through magnetron sputtering followed by a lift-off process upon the SiO₂ cladding trench window (process temperature <150 °C). Afterward" -> "deposited through magnetron sputtering followed by a lift-off process upon opening the SiO₂ cladding trench window (processing temperature <150 °C). Afterwards"

"to achieve optical loss equilibrium" -> "to balance optical losses on both arms"

Response:

We sincerely appreciate the reviewer for the helpful comment. We apologize for the grammatical errors in our previous manuscript. To address this issue, we have diligently rectified all the concerns raised by the reviewer. Moreover, we have entrusted a professional editing agency (AJE, Part of Springer nature) to enhance the overall readability of our manuscript. We have thoroughly revised our manuscript and the grammatic errors listed as example by the reviewer have been revised as outlined below:

To achieve this **goal**, the introduction of a silicon nitride **etch** stop layer at **select areas**, coupled with low-loss oxide trench **on the** waveguide surface, enables incorporation of various functional materials without disrupting the reliability of foundry-verified devices.

Introducing innovative materials for **functional** units **into** silicon photonics has become imperative to attain exceptional device performance and reduce power consumption.

The ongoing trend of integrating nonvolatile modulation materials such as charge-trapping materials, ferroelectric materials, and chalcogenide PCMs is crucial for lowering the static power consumption of reconfigurable photonic circuits.

Therefore, back-end-of-line integration of PCMs is desirable for commercial photonics platform-compatible nonvolatile devices.

Second, sequential deposition of 5-nm SiO₂ and 20-nm SiN was performed, followed by lithography and etching to fabricate the etch stop layer.

A PCM thin film was deposited through magnetron sputtering followed by a lift-off process upon opening the SiO₂ cladding trench window (processing temperature <150 °C). Afterward, the chip was annealed at 200 °C and 300 °C in a nitrogen atmosphere for 15 minutes to facilitate crystallization of Sb₂Se₃ and GeSbSe₄Te₁, respectively.

Reviewer #2 (Remarks to the Author):

The authors report on a technique that enables the monolithic BOEL integration of phase change materials. This platform/technique is labelled as "Zero Change" as it is fully compatible with standard silicon photonic processes and does not require modifications in their PDKs. This is a very important topic as the hybrid integration of other photonic materials to silicon chips is essential to enable functionalities such as amplification, phase shifting and attenuation, to cite a few. The value here resides is that the FOEL process does not need to be changed.

In this particular paper the authors report a technique that is based on etching down SiO₂ narrow trenches on the top of Si waveguide cores with a SiN etching stop layer that enables the safe post-deposition of other photonic materials. This technique is original to my understanding. The authors have successfully deposited two different chalcogenide PCMs (Sb₂Se₃ and Ge₂Sb₂Se₄Te) and demonstrated the operation in three applications (post-trimming, phase and amplitude modulation).

The paper reports a relevant progress over the state of the art and my suggestions and comments for minor revisions are focused towards improving some specific parts.

a) Regarding Fig 1(a) it is difficult to appreciate the limits of some of the regions. I believe this is because the colors that are chosen have little contrast. In particular I refer to the silicon and silica layers. Perhaps changing one of them can improve the visibility. Also in that figure the authors should specify the trenches limits

Response:

Many thanks to the reviewer for the valuable comment. Sorry for the inadequate contrast of the colors that were initially chosen. In response, we have made revisions to the flowchart, ensuring clear visibility of the trench in the revised figure. The revised figure has been added to our revised manuscript, and is presented below:

For Fig.1

b) when discussing the technique the authors indicate that a post-fabrication patterning will be required including annealing. How compatible would this be with a standard silicon photonics fabrication (i.e. would this be need to be carried in a completely different facility?)

Response:

Many thanks to the reviewer for the meaningful comment. The post-fabrication processes temperatures of phase change materials integration are all less than 300 °C and the flow could be compatible with standard silicon photonics fabrication. The device can be fabricated in the same facility as silicon photonic devices, provided that additional equipment for PCM sputtering, liftoff, and annealing processes is added to the process line. In recent years, the PCRAM industry has developed quite well, thereby the reliable large-scale deposition of phase change materials is feasible. Therefore, the backend integration of the PCM could be achieved in the same facility as long as those equipments are available.

In this work, we could not achieve the integration within the same facility. This is because the process line we employed is a pilot CMOS line where the incorporation of PCM is forbidden. Consequently, backend integration was conducted at the university fab facility. We are confident in the feasibility of wafer-scale backend integration of PCMs within a single facility in near future, and we are actively pursuing this objective.

c) The materials considered in this study are chalcogenide based and the authors mention that the technique can open the path to integrate "other excellent optoelectronic materials". however which materials are is not further discussed. can the authors elaborate more on this topic?

Response:

We extend our sincere gratitude to the reviewer for the comment. All optoelectronic materials with fabrication and annealing temperatures below 450 °C can be integrated into the silicon photonic chips employing the scheme we proposed. Typical materials encompass PCMs, quantum dots, barium titanate (BTO), lithium niobate (LiNbO_3), and electro-optic polymers.

In the future, photonic chips hold immense potential for applications in various fields including optical communication, bio-chemical sensing, optical computing, quantum computing, OPA-based LiDAR [R1]. Improved on-chip photonic components, such as near-zero-static-power-consumption phase shifters, optical amplifiers, high-speed modulators, and broadband detectors, are imperative for these applications. However, the current silicon-based optoelectronic integrated technology falls short in meeting these requirements. Hence, it is imperative to expedite the development of novel integrated photonic devices. An effective strategy involves combining silicon photonics with excellent optoelectronic materials such as PCMs, quantum dots[R2], barium titanate (BaTiO_3 , BTO)[R3], lithium niobate(LiNbO_3)[R4], electro-optic polymers[R5], and van der Waals materials[R6], to fulfill these demands. However, due to the incompatibility of these materials with conventional silicon photonics fabrication processes, it is challenging to integrate such materials. Hence, a back-end-of-line (BEOL) integration strategy is coveted. To address this challenge, we have developed a backend integration approach using SiN as an etch stop layer, which avoids the modification of PDKs. In our manuscript, we have demonstrated the integration of PCMs (SbSe and GSST), achieving low-static-power-consumption photonic devices for applications including post-trimming, phase modulation and amplitude modulation.

Moreover, to meet the requirements of future on-chip light amplification, we have attempted to integrate quantum dot material (approximately 100-nm PbS).

The PbS quantum dot (PbS QD) is integrated on top of the silicon waveguide via backend integration, and the fabricated device is shown in the inset of Fig.R1. After the patterning of the PbS QD, a photo luminescence measurement was conducted. The measured optical response under continuous exposure to a 1064-nm laser is shown in Fig.R1. Spatial light measurements reveal that the fluorescence of PbS was excited by the 2 mW laser. The obtained result is similar to our previous measurements conducted during the development of PbS QDs, suggesting that there was no significant deterioration of the PbS QD observed throughout the backend integration process. Although light amplification has not yet been achieved, it is expected to be achieved with further efforts involved.

We have conducted backend integration experiments with PbS QD. However, because the current device configuration is designed for PCM-based devices, the performance of these devices is suboptimal. We are now working on further iterative optimizations to improve the performance of these devices. We have added a discussion of the excellent optoelectronic properties of these materials to our revised manuscript, as follows:

To achieve high-performance hybrid integrated silicon photonic devices with novel optoelectronic materials, such as PCMs, quantum dots, barium titanate (BaTiO₃, BTO), lithium niobate (LiNbO₃), electro-optic polymers, and van der Waals materials, minimizing the distance between the materials and silicon waveguides is crucial.

The “near-zero-change” monolithic back-end-of-line integration platform could also pave the way for the integration of other excellent optoelectronic materials (including PCMs, quantum dots, BTO, LiNbO₃, electro-optic polymers, and van der Waals materials) into future silicon photonic PDKs.

Fig.R1 Raman spectroscopy measurement of 100-nm PbS deposited on a silicon waveguide. The inset is the device under test. The blue circle denotes the location of the continuous 1064-nm laser input, with a power of 2 mW.

d)When dealing with the push-pull MZI switch an insertion loss of 0.46 dB is reported. This is a quite high figure, especially considering an application to programmable

photonics where a target figure of 0.1-0.15 dB per gate is desired for scalable chips. can the authors comments on possible ways to reduce such losses?

Response:

Many thanks to the reviewer for the helpful comment. The insertion loss of the trimming unit is mainly due to the loss induced by the mode mismatch and phase transition of Sb_2Se_3 . Considering the mitigation of the mode mismatch loss between the $\text{Sb}_2\text{Se}_3/\text{Si}$ hybrid waveguide and silicon waveguide, designing the Sb_2Se_3 structure is an effective approach. As shown in Fig.R2, using a tapered hybrid waveguide as a transition region, combined with the deposition of Sb_2Se_3 solely on top of the silicon waveguide, can mitigate the insertion loss of the trimming unit. The simulated optical loss at 1550 nm indicates that the insertion loss is negligible when employing tapered Sb_2Se_3 (see Fig.R2 (b)). Therefore, the insertion loss can be mitigated by the geometric design of Sb_2Se_3 patterns.

The loss induced by the phase transition of Sb_2Se_3 is primarily attributed to scattering loss in its crystalline state. The scattering loss induced by crystalline Sb_2Se_3 is estimated by employing the approximated Mie theory (see Fig.R3)[R7]. For the $\text{Sb}_2\text{Se}_3/\text{Si}$ hybrid waveguide in our manuscript, the mode confinement in the Sb_2Se_3 film is usually 6%. The Sb_2Se_3 materials we developed exhibited a grain size of ~ 200 nm[R8]. The theoretical minimum scattering loss of the crystalline $\text{Sb}_2\text{Se}_3/\text{Si}$ hybrid waveguide is 0.0102 dB/ μm . Consequently, achieving low-loss phase shifts using the developed Sb_2Se_3 film poses significant challenges. However, the scattering loss in crystalline Sb_2Se_3 can be mitigated by reducing its grain size in the crystalline state. Although we have not yet achieved a low-loss optical switch, we carried out a theoretical analysis and elucidated an approach to mitigate the scattering loss in crystalline Sb_2Se_3 by reducing the grain size of the material.

In conclusion, the combination of Sb_2Se_3 pattern design and grain size reduction of Sb_2Se_3 material is expected to yield a low-loss nonvolatile phase shifter, thereby achieving a low-loss optical switch.

Fig.R2 The graded Sb_2Se_3 pattern effectively mitigates the insertion loss in the $\text{Sb}_2\text{Se}_3/\text{Si}$ hybrid waveguide, especially in crystalline Sb_2Se_3 . (a) Various hybrid waveguide configurations were explored, with Sb_2Se_3 selectively deposited on the surface of silicon waveguides. (b) The corresponding simulated insertion loss of different hybrid waveguide configurations.

Fig.R3 The scattering loss induced by grain boundaries in crystalline Sb_2Se_3 material estimated by approximated Mie theory.

e) In the case of the integration of SbSe in a MRR for multilevel phase modulation the authors report a phase shift of 0.3π . However for full functionality a 2π phase shift should be achieved. Can the authors comment?

Response:

Many thanks to the reviewer for the valuable comment. The determination of the 0.3π phase shifter in the MRR relies on the thickness (~ 25 nm) of Sb_2Se_3 and the length (~ 8 μm) of the $\text{Sb}_2\text{Se}_3/\text{Si}$ hybrid waveguide. The phase shift per unit length in an $\text{Sb}_2\text{Se}_3/\text{Si}$ hybrid waveguide, with a fixed width of Sb_2Se_3 , is dependent on the thickness of the Sb_2Se_3 layer. Increasing the length can improve the phase shift of a phase shifter, when the phase shift per unit length is fixed. Therefore, employing a thicker Sb_2Se_3 layer or a longer $\text{Sb}_2\text{Se}_3/\text{Si}$ hybrid waveguide can achieve a complete 2π phase shift. In our previous experiment, an $\sim 1.43\pi$ phase shift was achieved by employing a thicker $\text{Sb}_2\text{Se}_3/\text{Si}$ hybrid waveguide with a length of 16 μm (see Fig.R4). Since an excessive phase shift is unnecessary for implementing an MRR switch, we did not use this device to demonstrate the phase-modulated switch. Therefore, increasing the thickness of the Sb_2Se_3 layer or extending the length of phase shifter represents a potential strategy for achieving a complete 2π phase shift.

Fig.R4 Measured optical response of an MRR incorporating a thicker Sb_2Se_3 layer on a silicon waveguide with a length of 16 μm .

f)I have real problems in reading the figures as the fonts are very small. Perhaps the authors could consider reducing the number of subfigures in the main document and transferring some to the supplementary materials.

Response:

Many thanks to the reviewer for the helpful comment. Sorry for the small font size used in the figures. To address this issue, we have increased the font size of the figures in our revised manuscript. Moreover, some of the figures have been transferred to the supplementary materials. The revised figures have been updated in our revised manuscript, as follows:

For Fig.2

■ Si WG
 ■ Trimming Unit
 ■ Balanced Loss
 ■ N++
 ■ P++
 ■ Metal

(a)

(b)

(c)

(d)

(e)

(f)

(g)

For Fig.3

For Fig.4

For the figures (inset and subfigures) moved to Supplementary information

For SI.5

g) A final discussion on reliability derived from the use of pulsed techniques and material transition from amorphous to crystalline would be useful to the reader.

Response:

Many thanks to the reviewer for the valuable comment. The achievement of partial crystallization, i.e., the desired arbitrary state configuration, can be realized through two distinct approaches. One approach necessitates only two pulses, while the other requires multiple pulses. In terms of reliability of multilevel switching during gradual crystallization, the approach employing multiple pulses proves to be more reliable.

For the first one, a pulse with short duration and high amplitude was initially employed, followed by another specific pulse to switch to the desired intermediate state. The first pulse is typically employed to induce amorphization. The second pulse determines the distinct intermediate states. Consequently, diverse state configurations can be attained by modifying the parameters of the second pulse. An arbitrary configuration, as illustrated in Fig.R5, was achieved through the application of pulses with varying amplitudes (durations) resulting in a total of 7 (11) repeatable and distinguishable states. However, achieving high resolution solely through the application of two pulses is challenging due to the inherent random nucleation during crystallization.

The desired intermediate state can be progressively approximated by fine-tuning in conjunction with continuous feedback monitoring. A reliable intermediate state can be achieved by applying a single pulse for amorphization, followed by multiple pulses for gradual crystallization. As shown in Fig.R6(a), >7-bit multilevel switching was achieved by applying pulses of various durations with a fixed amplitude (PVDFFA). Moreover, the quasi-continuous crystallization has been demonstrated by employing PVDFFA (Fig.R6(b)). Therefore, the reliable configuration of arbitrary states can be achieved by combining multiple pulses and feedback monitoring.

We have added a discussion on the reliability of multilevel switching by employing varied pulses in our revised manuscript. All changes are shown below:

By applying pulses with various amplitudes (various durations), a total of 7 (11) distinguishable states were achieved (see Fig. 4(g) and (h)). Achieving high resolution solely through the application of two pulses is challenging due to the inherent random nucleation during crystallization. To obtain a reliable arbitrary state configuration, combining fine-tuning with feedback monitoring is a promising solution. A reliable intermediate state can be achieved by applying a single pulse for amorphization, followed by multiple pulses for gradual crystallization. Although a specific state

configuration requires multiple pulses, this approach is anticipated to achieve precise multilevel switching and even quasi-continuous configurations.

Fig.R5 Measured arbitrary configuration of the GSS4T1-based photonic attenuator. (a) The measured transmission change (ΔT) by applying pulses with varying amplitudes. Seven states are distinguishable as indicated by the shaded regions. (b) The measured transmission changes by applying pulses of varying duration. There are 11 distinguishable states.

Fig.R6 The gradual crystallization of PCM. (a) Gradual crystallization of the GSS4T1-based photonic attenuator. (b) Measured quasi-continuous crystallization of the Sb_2Se_3 -based MRR switch. The transmittance change (ΔT) was measured at 1551.513 nm.

References

- R1. Zhou, Z., et al., *Prospects and applications of on-chip lasers*. eLight, 2023. **3**(1): p. 1.
- R2. Duan, J., et al., *Dynamic and nonlinear properties of epitaxial quantum dot lasers on silicon for isolator-free integration*. Photonics Research, 2019. **7**(11): p. 1222-1228.
- R3. Abel, S., et al., *Large Pockels effect in micro- and nanostructured barium titanate integrated on silicon*. Nature Materials, 2019. **18**(1): p. 42-47.
- R4. He, M., et al., *High-performance hybrid silicon and lithium niobate Mach-Zehnder modulators for 100 Gbit s⁻¹ and beyond*. Nature Photonics, 2019. **13**(5): p. 359-364.
- R5. Melikyan, A., et al., *High-speed plasmonic phase modulators*. Nature Photonics, 2014. **8**(3): p. 229-233.
- R6. Ma, H., et al., *Enhanced Light-Tellurium Interaction through Evanescent Wave Coupling for High Speed Mid-Infrared Photodetection*. Advanced Optical Materials, 2022. **10**(23): p. 2201443.
- R7. Wiscombe, W.J., *Improved Mie scattering algorithms*. Applied Optics, 1980. **19**(9): p. 1505-1509.

- R8. Lei, K., et al., *Magnetron-sputtered and thermal-evaporated low-loss Sb-Se phase-change films in non-volatile integrated photonics*. *Optical Materials Express*, 2022. **12**(7): p. 2815-2823.

REVIEWERS' COMMENTS

Reviewer #1 (Remarks to the Author):

I would like to thank the authors for making the detailed revision. Most of my questions have been nicely addressed, with the only exception being my comment on "zero-change". I find the term "near-zero-change" coined by the authors oxymoronically amusing and I suggest against using it. Instead, the authors may emphasize on the "foundry-manufactured" aspect as the key point. The title of the paper for example can be revised to "Monolithic back-end-of-line integration of phase change materials into foundry-manufactured silicon photonics". The conclusion can be changed to "The validated foundry-compatible, monolithic back-end-of-line integration platform will also ...". Similar changes should be made to other references to "near-zero-change".

Other than these minor changes, this manuscript is ready for publication.

Reviewer #2 (Remarks to the Author):

The authors have addressed adequately all my concerns providing a very detailed discussion in several parts of their reply. I am happy to recommend the publication of the paper.

Response to Reviewer Comments NCOMMS-23-47181-A

REVIEWER COMMENTS

Reviewer #1 (Remarks to the Author):

I would like to thank the authors for making the detailed revision. Most of my questions have been nicely addressed, with the only exception being my comment on "zero-change". I find the term "near-zero-change" coined by the authors oxymoronically amusing and I suggest against using it. Instead, the authors may emphasize on the "foundry-manufactured" aspect as the key point. The title of the paper for example can be revised to "Monolithic back-end-of-line integration of phase change materials into foundry-manufactured silicon photonics". The conclusion can be changed to "The validated foundry-compatible, monolithic back-end-of-line integration platform will also ..." Similar changes should be made to other references to "near-zero-change".

Other than these minor changes, this manuscript is ready for publication.

Response:

Thank you very much for the valuable suggestion provided by the reviewer. We greatly appreciate the feedback and have taken it into consideration. As a result, we have made further revisions to the title of the manuscript and have also modified certain expressions in the abstract, introduction, and discussion sections. The revised contents are listed below, with the modifications highlighted in red:

For the title:

Monolithic back-end-of-line integration of phase change materials into **foundry-manufactured** silicon photonics

For the abstract:

This foundry-compatible platform could open up the possibility of integrating other excellent optoelectronic materials into future silicon photonic process design kits.

For the introduction:

In this paper, a **foundry-compatible** platform for monolithic back-end-of-line integration **was demonstrated, facilitating large-scale integration of PCM-based photonic devices.**

For the discussion:

In this study, we presented a platform for monolithic back-end-of-line integration of PCMs into **foundry-manufactured** silicon photonics.

The **validated foundry-compatible,** monolithic back-end-of-line integration platform **will** also pave the way for the integration of other excellent optoelectronic materials (including PCMs, quantum dots, BTO, LiNbO₃, electro-optic polymers, and van der Waals materials) into future silicon photonic PDKs.

Reviewer #2 (Remarks to the Author):

The authors have addressed adequately all my concerns providing a very detailed discussion in several parts of their reply. I am happy to recommend the publication of the paper.

Response:

We appreciate it very much that the reviewer recommended our paper for publication. We are thankful for all the advice and comments given by the reviewer, which has helped us improve the quality of our work.